# p190RhoGAP proteins contain pseudoGTPase domains

Amy L. Stiegler[1] & Titus J. Boggon[1,2,3]

The two p190RhoGAP proteins, p190RhoGAP-A and -B, are key regulators of Rho GTPase signaling and are essential for actin cytoskeletal structure and contractility. Here we report the discovery of two evolutionarily conserved GTPase-like domains located in the 'middle domain', previously thought to be unstructured. Deletion of these domains reduces RhoGAP activity. Crystal structures, MANT-GTPγS binding, thermal denaturation, biochemical assays and sequence homology analysis all strongly support defects in nucleotide-binding activity. Analysis of p190RhoGAP proteins therefore indicates the presence of two previously unidentified domains which represent an emerging group of pseudoenzymes, the pseudoGTPases.

[1] Department of Pharmacology, Yale University School of Medicine, 333 Cedar Street, New Haven, CT 06520, USA. [2] Department of Molecular Biophysics and Biochemistry, Yale University School of Medicine, 333 Cedar Street, New Haven, CT 06520, USA. [3] Yale Cancer Center, Yale University School of Medicine, 333 Cedar Street, New Haven, CT 06520, USA. Correspondence and requests for materials should be addressed to T.J.B. (email: titus.boggon@yale.edu)

In recent years, 'pseudoenzymes' have shown key roles for enzymatic folds in regulation and control of signal transduction pathways using noncatalytic methods. For example, the kinase family has an extensive 'pseudo-' group with members that adopt the kinase fold but are degraded in one or more of the consensus motifs required for adenosine triphosphate (ATP) binding and catalysis[1]. Instead, the pseudokinases are implicated in scaffolding/adaptor roles in signal transduction. The Ras superfamily of small GTPases consists of over 150 proteins that act as molecular switches in broad and diverse cellular pathways and processes[2], but relatively few pseudoGTPases have been discovered. GTPases bind to guanosine triphosphate (GTP), hydrolyze γ-phosphate, release guanosine diphosphate (GDP) and then re-bind GTP, a process termed 'GTPase cycling'. This cycling, and consequent signal transduction, is regulated by GTPase activating proteins (GAP) (for GTP hydrolysis) and guanine nucleotide exchange factors (GEF) (for GDP release). The five Ras superfamily subgroups (Ras, Rho, Rab, Ran and Arf) each contain five highly conserved sequence motifs, termed 'G-motifs', required for nucleotide-binding and catalytic activity[2]. Like pseudokinases, pseudoGTPases by definition would consist of a GTPase fold lacking one or more of these G motifs.

Small GTPases in the Rho subgroup (including RhoA, Cdc42 and Rac1) mediate signaling from the cell membrane to the actin cytoskeleton and play key roles in cellular functions such as adhesion, migration and cytokinesis, and in disease-associated processes such as cell growth and metastasis in cancer[3]. The p190RhoGAP proteins, p190RhoGAP-A (ARHGAP35) and p190RhoGAP-B (ARHGAP5)[4–6], are key regulators of Rho GTP hydrolysis and are highly important for maintenance of proper Rho signaling. They share over 50% sequence identity and a domain organization containing a GTP-binding GTPase domain, four FF domains and a C-terminal GAP domain. An ~700 amino-acid stretch between the FF and GAP domains is termed the 'middle domain'[7] (Fig. 1a), and is thought to be unstructured.

In the current study, we set out to gain a more comprehensive description of the domain architecture of the p190RhoGAP proteins in order to better understand their functions. We identify two GTPase-like folds in the middle domain of both p190RhoGAP-A and -B which we term pG1 and pG2, and show that they are important for GAP activity toward RhoA. Crystal structures of pG1 reveal similarities to the Ras-like GTPase superfamily. However, conserved GTPase motifs are notably absent from both domains, and binding assays support a lack of nucleotide-binding activity. Thus, we classify p190RhoGAP pG1 and pG2 as psuedoGTPases.

## Results

### Two GTPase-like folds are predicted in p190RhoGAP.

Although our initial analyses indicate that the middle domains of p190RhoGAP proteins indeed contain unstructured regions, secondary structure prediction algorithms unexpectedly indicate extensive order for the amino-terminal half of the middle domain, corresponding to residues 550–960 of p190RhoGAP-A (human) (Fig. 1a). Homology detection within this region strongly predicts two distinct domains encompassing residues 592–767 and 779–950 of human p190RhoGAP-A. Despite sequence identities below 20%, both domains have very high predicted homology (probability scores over 99% and E-values of $6 \times 10^{-19}$) and matched secondary structure with members of the Ras superfamily of small GTPases (Supplementary Fig. 1 and Supplementary Table 1). Consistently, these tandem GTPase-like domains are predicted by homology detection in human p190RhoGAP-B, in p190RhoGAP-A and -B from diverse vertebrate species, and in the single p190RhoGAP protein from lower species including *Drosophila melanogaster* and the sponge *Amphimedon queenslandica* (Supplementary Table 1). These domains, however, show low sequence identity with the known N-terminal GTPase in p190RhoGAP (12 and 21% in p190RhoGAP-A). Overall, these putative GTPase-like domains appear to be conserved across evolution in p190RhoGAP proteins. We term these domains 'pG1' and 'pG2' (Fig. 1b).

### Crystal structure of pG1 domains from p190RhoGAP-A and -B.

We obtained crystals of the p190RhoGAP-A and p190RhoGAP-B pG1 domains that diffract to 1.9 and 2.6 Å resolution, respectively (Table 1 and Supplementary Fig. 2a). The resulting crystal structures clearly reveal a small GTPase fold in both cases, with a central 6-stranded β-sheet surrounded by four α-helices[8] (Fig. 2a and Supplementary Fig. 2b, c). The root mean square deviation (r.m.s.d.) between the two pG1 structures is 1.8 Å over 128 $C_{\alpha}$s (Fig. 2b), and 2.7 Å (135 $C_{\alpha}$s) between the prototypical small GTPase H-Ras and p190RhoGAP-A pG1 (Fig. 2c and Supplementary Fig. 2b, c). Importantly, neither pG1 domain crystal structure contains any electron density that could be interpreted as bound nucleotide.

Nucleotide-binding and catalytic activity of small GTPases requires correct positioning of the G motifs termed G1 (phosphate-binding P-loop), G2 (Switch I), G3 (Switch II), G4 and G5[2]. All five G motifs are degraded in p190RhoGAP pG1 (Fig. 3a). In canonical GTPases, the P-loop (G1) forms a cavity to bind phosphates of GDP/GTP. In p190RhoGAP, helix α1 (which follows the P-loop) is extended and sterically occludes the phosphate-binding cavity (Fig. 3a, b). Switch I (G2) is normally

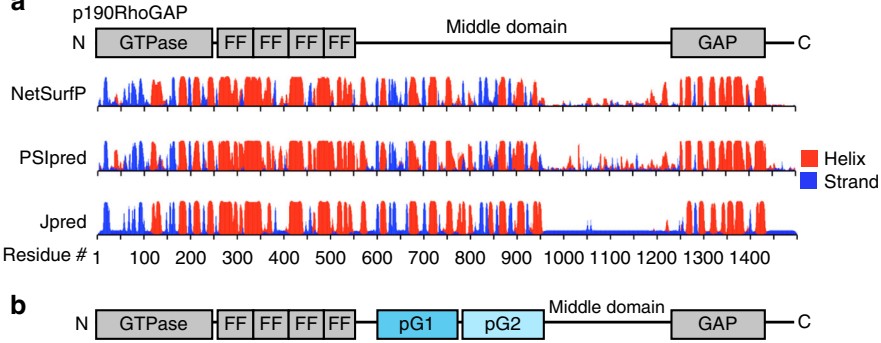

**Fig. 1** p190RhoGAP proteins contain pseudoGTPase domains. **a** Current domain assignment for p190RhoGAP proteins (*top*). Secondary structure predictions of p190RhoGAP-A from humans (*below*) shown as probability of secondary structure (*y* axis) against residue number (*x* axis). Predicted helices in *red* and strands in *blue*. **b** Updated domain assignment for p190RhoGAP proteins

**Table 1 Data collection and refinement statistics**

| Data collection | p190RhoGAP-A pG1 | p190RhoGAP-B pG1 |
|---|---|---|
| PDB accession code | 5U4U | 5U4V |
| Species | Xenopus laevis | Homo sapiens |
| X-ray source | APS ID-24-E | APS ID-24-E |
| Number of crystals | 1 | 1 |
| Wavelength (Å) | 0.97915 | 0.97915 |
| Space group | $P6_5 22$ | $P4_3$ |
| Cell dimensions $a,b,c$ (Å) | 62.9, 62.9, 161.0 | 40.8, 40.8, 95.2 |
| Cell dimensions $\alpha$, $\beta$, $\gamma$ (°) | 90, 90, 120 | 90, 90, 90 |
| Resolution range (Å)[a] | 50–1.9 (1.97–1.90) | 50–2.6 (2.69–2.60) |
| Unique reflections | 15,922 | 4908 |
| Multiplicity[a] | 8.9 (7.7) | 4.1 (4.2) |
| Completeness (%)[a] | 99.7 (99.4) | 99.9 (100) |
| $R_{pim}$(%)[a] | 7.9 (75.2) | 6.6 (53.7) |
| $<I>/<\sigma I>$[a] | 11.7 (1.5) | 11.6 (1.8) |
| Wilson $B$-factor | 22.4 | 54.3 |
| | | |
| *Refinement* | | |
| Resolution range (Å)[a] | 45.13–1.90 (2.02–1.90) | 40.76–2.60 (2.67–2.60) |
| $R_{factor}$ (%)[a] | 18.1 (29.4) | 24.4 (20.8) |
| Free $R_{factor}$ (%)[a] | 21.7 (39.2) | 27.4 (33.0) |
| No. of free $R$ reflections[a] | 802 (117) | 240 (14) |
| Free $R$ reflections (%)[a] | 5.1 (4.9) | 5.0 (3.3) |
| Residues built | 157 total: 585–677, 688–700, 708–733, 738–762 (585–586 are vector-derived) | 135 total: 594–639, 655–685, 691–702, 710–733, 739–761 |
| No. of non-hydrogen atoms | 1391 | 1080 |
| No. of protein atoms | 1293 | 1075 |
| No. of water molecules | 81 | 5 |
| No. of solvent molecules | $Na^{2+}$ (3), malonate (2) | – |
| Solvent atoms | 17 | – |
| | | |
| *Model quality* | | |
| RMSD bond lengths (Å) | 0.007 | 0.012 |
| RMSD bond angles (°) | 0.95 | 1.4 |
| Overall $B$ (all atoms) (Å$^2$) | 33.7 | 58.7 |
| $B$ (Å$^2$) (protein) | 33.0 | 58.8 |
| $B$ (Å$^2$) (solvent) | 42.7 | 39.4 |
| $B$(Å$^2$) (ligand) | 42.3 | – |
| Ramachandran plot (%): favored/allowed/outliers | 98.1/1.9/0.0 | 99.2/0.8/0.0 |
| Mol Probity score (percentile) | 1.2 (99%) | 1.6 (99%) |

[a]Numbers in parentheses indicate the highest resolution shell

associated with regulated effector binding to small GTPases, undergoes conformational changes during nucleotide cycling and contains a conserved threonine that contacts the GTP γ-phosphate and $Mg^{2+}$ (Fig. 3a). Switch I (G2) is completely absent in p190RhoGAP pG1 due to an 11-residue deletion (Fig. 3a and Supplementary Fig. 2b, c). Switch II (G2) is also degraded in p190RhoGAP pG1, and places a glutamate side chain (Glu636) where the γ-phosphate of GTP binds to canonical small GTPases (Fig. 3b). The G4 and G5 motifs are disordered in our structures, but their sequence divergence (Fig. 3a) suggests degraded ability to bind guanine base, and in our structures the guanosine-binding site appears to be sterically hindered by the side chains of Glu606 and Arg739, which form a predicted salt bridge with one another (Fig. 3b). Together, the substantially degraded G motifs in the p190RhoGAP pG1 domain indicate an inability to bind nucleotide.

**pG1 is not a nucleotide-binding domain**. To test nucleotide-binding directly, we performed binding assays using the fluorescent nonhyhdrolyzable GTP analog MANT-GTPγS (2′/3′-O-(N-Methyl-anthraniloyl)-guanosine-5′-(γ-thio)-triphosphate). We tested binding in the presence of magnesium, and in both the presence and absence of EDTA, since our positive control protein Rac1 requires EDTA for strong MANT-GTPγS binding[9].

Consistent with our hypothesis that pG1 does not bind nucleotide, we find no change in fluorescence of MANT-GTPγS upon addition of either p190RhoGAP-A or -B pG1 in these conditions[9] (Fig. 4a). We find that this lack of nucleotide binding for pG1 domains is evolutionarily conserved across diverse species, including pG1 from the single p190RhoGAP of D. melanogaster (Fig. 4b). We further tested MANT-GTPγS binding in the presence of a panel of atypical divalent cations including $Mn^{2+}$, $Ca^{2+}$ and $Zn^{2+}$ and find that pG1 does not bind MANT-GTPγS in the presence of these atypical divalent cations (Fig. 4c).

We next employed a thermal shift assay[10] to investigate potential nucleotide and divalent cation binding by pG1. We find that the melting temperatures ($T_m$) of pG1 from p190RhoGAP-A and -B are unchanged in the presence of divalent cation, nucleotide or both. This contrasts with Rac1 which exhibits a positive shift in $T_m$ in the presence of ligands, reflective of direct binding (Fig. 4d–i). Consistent with our biochemical studies, co-crystallography of p190RhoGAP-A pG1 with GTPγS and $Mg^{2+}$ yields no conformational differences and no electron density corresponding to bound nucleotide. Taken together, these studies establish that the isolated pG1 domain of p190RhoGAP is not a nucleotide-binding domain.

**p190RhoGAP pG2 is a predicted pseudoGTPase**. Due to a current lack of structural information for the p190RhoGAP pG2

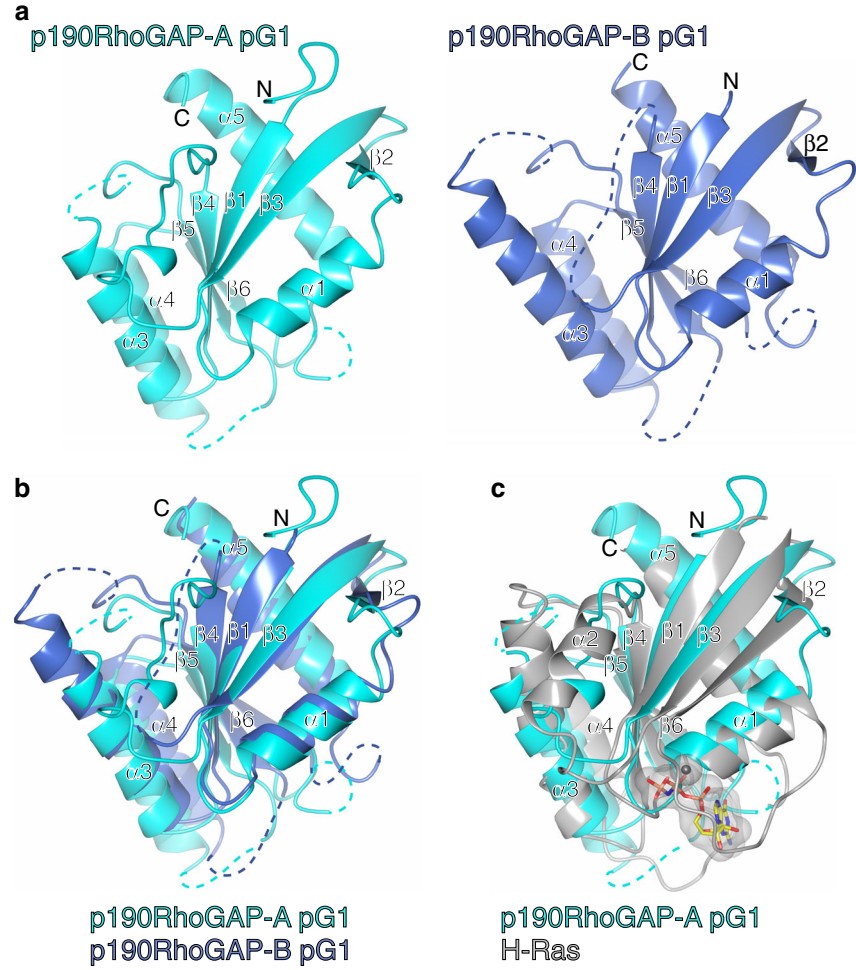

**Fig. 2** Structural analysis of p190RhoGAP pseudoGTPase domain pG1. **a** Crystal structures of pG1 for p190RhoGAP-A (*left, cyan*) and p190RhoGAP-B (*right, blue*). Secondary structure and N and C termini are labeled. Unmodeled loops are indicated by a *dashed line*. **b** Superposition of p190RhoGAP-A and -B pG1 domains. **c** Superposition of p190RhoGAP-A pG1 domain with GTP-analog bound H-Ras in *gray*; PDB ID: 5P21[56]

domain, we employed sequence alignment and homology prediction. Similar to pG1, the five G motifs of pG2 are highly degraded (Supplementary Fig. 3 and Supplementary Table 1). From these alignments it is evident that the predicted G motifs in pG2 are degraded from the consensus sequences of active small GTPases like Rab8, Ras3 and Rad.

**Rhotekin-binding assay to assess active RhoA**. To assess a potential role for pG1–pG2 in p190RhoGAP activity toward RhoA, we performed Rhotekin pulldown assays comparing levels of active RhoA at varying expression levels of p190RhoGAP-A wild-type or a deletion mutant lacking pG1–pG2 (ΔpG1–pG2) (Fig. 5). We observe that p190RhoGAP-A ΔpG1–pG2 leads to higher levels of active RhoA compared to wild-type p190Rho-GAP-A (Fig. 5a, b). When data are grouped into bins of similar expression levels of p190RhoGAP (1–10%, 10–25%, 25–50% and 50–100% of maximum expression in each experiment), we find that a significantly higher level of active RhoA is found in cells expressing mutant p190RhoGAP-A compared to wild-type p190RhoGAP-A (Fig. 5b, c). Thus, removal of pG1–pG2 from p190RhoGAP-A appears to decrease its activity toward RhoA in this system. This raises the exciting possibility that pG1–pG2 is important for full p190RhoGAP activity. This observation may be due to biological effects such as differences in p190RhoGAP localization, as it has recently been suggested that regions near pG1–pG2 drive p190RhoGAP localization to membrane

protrusions[11]. Additionally, reported binding partners of p190RhoGAP such as Rnd3 may bind the pG1–pG2 region and affect p190RhoGAP activity[12]. Interestingly, a conserved surface on p190RhoGAP-A pG1 is observed (Supplementary Fig. 4) potentially suggesting a protein interaction surface.

## Discussion

The pseudokinases represent a paradigm for analysis of other pseudoenzyme folds[1]. Similar to the G motifs of small GTPases, canonical kinases contain consensus motifs responsible for catalysis and ATP binding[13]. Pseudokinases are degraded in one or more of these motifs, and can be further classified based on whether they are: (i) unable to bind nucleotide, (ii) able to bind nucleotide but have no catalytic activity or whether they (iii) retain catalytic activity (albeit low in some cases)[14]. There are numerous examples of pseudokinases that fall into this classification system (Table 2)[15, 16]. These classifications have important implications for function of the pseudokinases, with the non-enzymatic groups thought to assume scaffolding/adaptor roles in signal transduction[17].

The pseudoGTPases can be classified in a similar manner. The existence of a pseudoGTPase subgroup unable to bind nucleotide (class i) is supported by our findings for p190RhoGAP, and by previous studies of the human centromere protein M (CENP-M)[18] and a fungal dynein motor protein light intermediate chain (LIC)[19] (Supplementary Table 2 and Table 2).

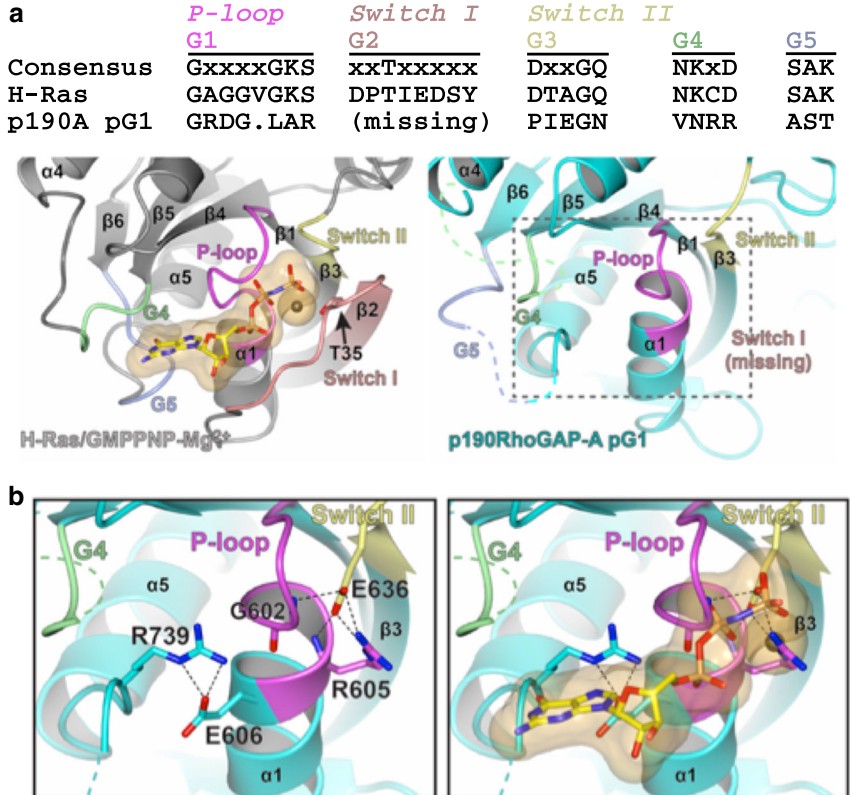

**Fig. 3** Analysis of the degraded nucleotide binding pocket of p190RhoGAP pG1. **a** Alignment (*top*) and structural comparison (*bottom*) of conserved G motifs for H-Ras (PDB ID: 5P21[56]) and p190RhoGAP-A pG1 (p190A). Conserved threonine T35 for H-Ras is shown. *Dashed box* indicates region shown in **b**. **b** Close-up of the disrupted nucleotide-binding site of p190RhoGAP-A pG1 domain (*left*) and additional space filling model indicating the location of GTP analog when H-Ras is superposed onto the structure (*right*)

Likewise, a pseudoGTPase subgroup able to bind nucleotide but with no catalytic activity (class ii) contains members of the Rnd family that have degraded sequences near the Switch II motif (Supplementary Table 2) and permanently bind GTP but do not catalyze hydrolysis[20] (Table 2). Lastly, a pseudoGTPase subgroup that retains catalytic activity (class iii) includes the RGK family, which can catalyze hydrolysis but diverges at several G-motif sequences[21] (Table 2 and Supplementary Table 2). We note that the G protein-like domain of the AGAP proteins (subgroup of the ArfGAP family) also lacks several conserved G motifs; however, reports of nucleotide binding are controversial[22]. Similar to pseudokinases, pseudoGTPases also exist across phylogenetic groups; for example, EhRabX3, a protein in the parasitic protozoan *Entamoeba hystolytica* contains a GTPase domain lacking multiple G motifs[23], and the bacterial CheY response regulator receiver superfamily comprises a small GTPase fold with degraded G motifs and no nucleotide binding[24, 25]. Therefore, our discovery of evolutionarily conserved pseudoGTPase domains in p190RhoGAP illustrates that, similar to pseudokinases, pseudoGTPases exist in mammalian cytosolic signaling cascades, and represent a growing class of pseudoenzymes with diverse nucleotide-binding, enzymatic activity and physiological functions. As with recently discovered pseudophosphatases, pseudoproteases and pseudodeubiquitinases[26, 27], understanding the function of pseudoGTPases will be an important future challenge with new mechanistic lessons in signal transduction.

## Methods

**Bioinformatics**. Domain assignment searches on full-length p190RhoGAP proteins (p190RhoGAP-A from *Homo sapiens*, UniProt Q9NRY4; and p190RhoGAP-B from *Homo sapiens*, UniProt Q13017) were conducted using the National Center for Biotechnology Information (NCBI) Blast/CDD[28] and InterPro[29]. Secondary

structure predictions were also conducted on full-length protein sequences using NetSurfP[30], PSIPRED[31] and JPred[32]. Primary sequence alignments were carried out in CLUSTALO[33]. Residues 550–960 of *Homo sapiens* p190RhoGAP-A, 590–950 of *Homo sapiens* p190RhoGAP-B, 593–927 of *D. melanogaster* p190RhoGAP (UniProt ID: Q9VX32) and 610–950 of sponge p190RhoGAP (*A. queenslandica*, NCBI Reference sequence XP_003385690.2) were submitted for homology detection used the HHpred server[34, 35]. Structural similarity searches were conducted using the Dali server[36]. All structural figures generated using CCP4MG[37].

**Expression constructs**. The complementary DNA (cDNA) encoding full-length *Rattus norvegicus* (rat) p190RhoGAP-A (ARHGAP35) protein (NCBI Reference Sequence: NP_001258061.1, UniP rotA0A0G2KB46) was inserted into a modified pCDNA-3.1 (Invitrogen) vector containing an N-terminal Flag tag. The full-length cDNA was used as a PCR template to amplify the predicted pG1 (residues 592–767) and pG2 (residues 766–958) regions. Similarly, regions encoding pG1 (590–763) or pG2 (764–954) of *Homo sapiens* (human) p190RhoGAP-B (ARHGAP5) (UniProt ID: Q13017) were generated by PCR using full-length p190RhoGAP-B cDNA as template. Codon-optimized synthetic cDNAs (Supplementary Table 3) encoding the pG1 region from *Gallus gallus* (chicken) p190RhoGAP-A (UniProt ID: A0A1D5P6Q7, residues 592–764, 90% identical to rat p190RhoGAP-A), *Xenopus laevis* (frog) p190RhoGAP-A (UniProt ID: Q6NU25, residues 587–762, 89% identical to rat p190RhoGAP-A), *Danio rerio* (zebrafish) p190RhoGAP-A (UniProt ID: F1R0X6, residues 592–772, 69% identical to rat p190RhoGAP-A) and *D. melanogaster* p190RhoGAP, which contains a single gene for p190 (UniProt ID: Q9VX32, residues 593–753, 22% identical to rat p190RhoGAP-A), were purchased from GenScript. All cDNA fragments encoding the pG1 and/or pG2 domains of p190RhoGAP-A and -B were inserted into a modified pET vector containing an N-terminal hexahistidine (His₆) tag followed by a recognition sequence for tobacco etch virus (TEV) protease for expression in *Escherichia coli*. Human Rac1 (UniProt ID: P63000) residues 2–177 was inserted into the pET28a plasmid for expression in *E. coli* as a His-tagged protein[9]. GST-RBD (Rhotekin) was a gift from Martin Schwartz (Addgene plasmid no. 15247)[38]. Mutant constructs were generated with QuikChange Lightning Site-Directed mutagenesis kit (Agilent). The sequences of primers used in this study are listed in Supplementary Table 4.

**Protein expression and purification**. His₆-tagged p190RhoGAP-A and -B pG1 proteins, and His-Rac1, were expressed in BL21 (DE3) cells (Millipore Sigma) or

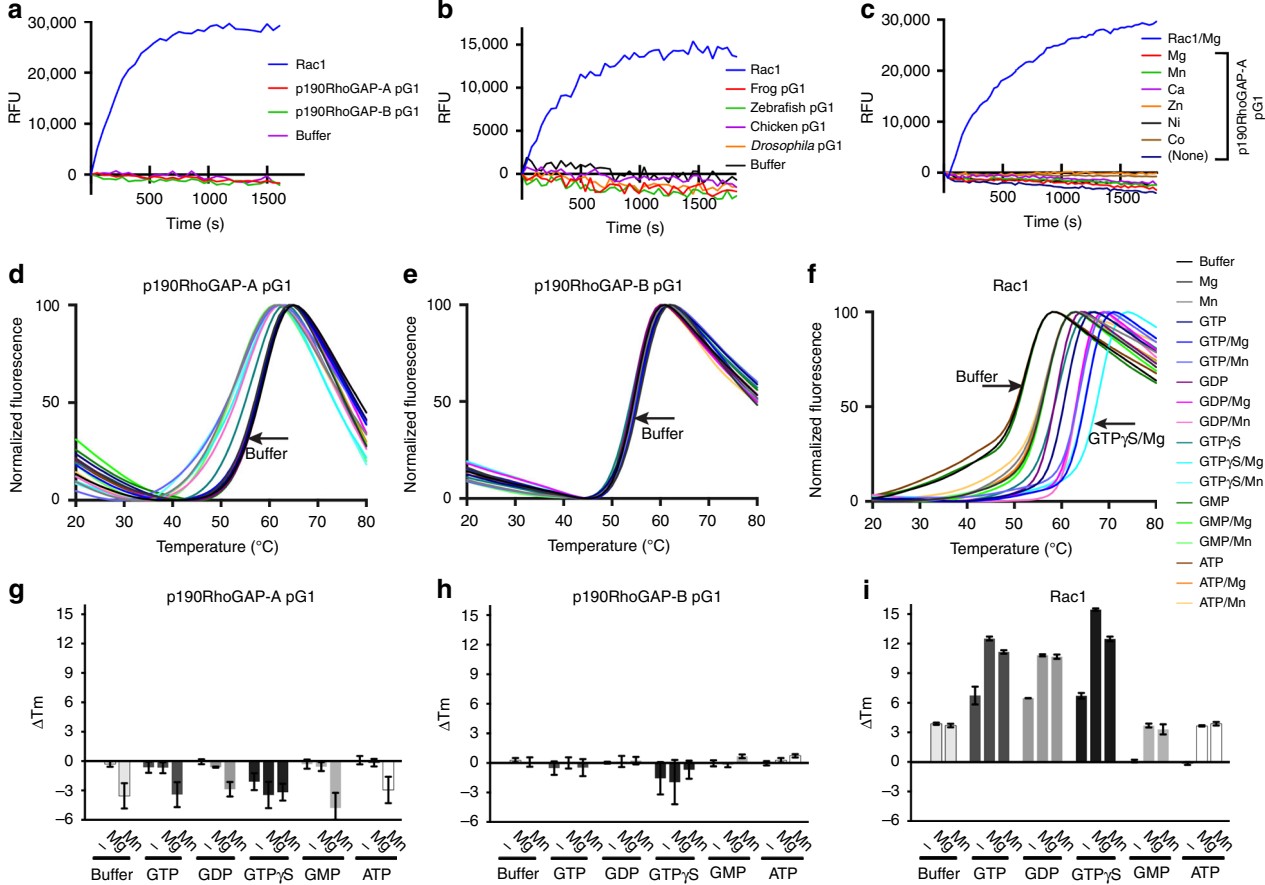

**Fig. 4** Assessment of nucleotide binding by p190RhoGAP pG1. **a–c** Fluorescence of MANT-GTPγS monitored over time upon addition of proteins as indicated. **a** pG1 from 190RhoGAP-A or -B, or Rac1 as a control, in the presence of 20 mM EDTA plus 10 mM MgCl2. **b** p190RhoGAP-A pG1 of *Xenopus laevis* (frog), *Gallus gallus* (chicken) and *Danio rerio* (zebrafish) or the single p190RhoGAP of *Drosophila melanogaster*. **c** p190RhoGAP-A pG1 in the presence of different divalent cations. Rac1 in $Mg^{2+}$ is included as a positive control. Uncorrected for photobleaching. **d–i** Thermal shift assay. **d–f** Thermal denaturation curves (from a representative experiment) of pG1 domains or Rac1 in the presence of divalent cation, nucleotide or both as listed in **f**. Buffer alone curves (*black*) are labeled with *arrows*. **g–i** Histograms of melting temperature changes ($\Delta T_m$) compared to buffer alone, determined by fitting the thermal denaturation curves (**d–f**) to a sigmoidal model. *Error bars* indicate s.e.m. ($n=3$). Positive shifts indicate a stabilization in the presence of ligand

Rosetta (DE3) cells (Millipore Sigma) by induction with 0.2–0.5 mM isopropyl β-D-thiogalactopyranoside (IPTG) overnight at 16 °C. Cells were harvested by centrifugation at 2,000×g and lysed in nickel binding buffer (50 mM HEPES pH 7.3, 500 mM NaCl, 5 mM imidazole) by addition of lysozyme followed by freeze/thaw cycles and sonication. Lysates were clarified by centrifugation at 5000×g for 1 h and filtration, and applied to nickel beads for affinity purification (Ni Sepharose 6 Fast Flow, GE Healthcare). Following elution of bound proteins by increasing concentrations of imidazole in nickel-binding buffer, the His6 tag was removed from p190RhoGAP proteins by incubation with TEV protease overnight during dialysis against buffer containing 20 mM Tris pH 7.5 and 150 mM NaCl. The cleavage reaction was then flowed over a nickel affinity column (HisTrap Fast Flow, GE Healthcare) to remove the His6 tags, uncleaved His6-tagged protein and the His6-tagged TEV protease. The flow-through containing untagged p190RhoGAP proteins was concentrated in a centrifugal filter (Amicon Ultra, Millipore Sigma) and applied to size exclusion chromatography (Superdex 75 prep grade, GE Healthcare) in 20 mM Tris pH 7.5 and 150 mM NaCl. We were unable to express soluble protein for pG2 or tandem pG1–pG2 for either p190RhoGAP-A or -B using either *E. coli*, insect cell or mammalian expression systems. His6-tagged wild-type human Rac1 was eluted from nickel beads with increasing concentrations of imidazole in nickel-binding buffer, loaded onto a size exclusion chromatography column (Superdex 75 prep grade, GE Healthcare) in 20 mM Tris pH 7.5 and 150 mM NaCl and 1 mM dithiothreitol (DTT) and fractions pooled and concentrated in a centrifugal filter (Amicon Ultra, Millipore Sigma). GST-RBD (Rhotekin) was expressed in BL21 (DE3) cells (Millipore Sigma) by induction with 0.5 mM IPTG overnight at 16 °C. Cells were harvested by centrifugation at 2.000×g and lysed in lysis buffer (20 mM Tris pH 7.5, 150 mM NaCl, 1 mM DTT, 5 mM MgCl2, 0.1 M phenylmethylsulfonyl fluoride (PMSF) and Roche cOmplete EDTA-Free protease inhibitor tablet) by addition of lysozyme followed by freeze/thaw cycles and sonication. Lysates were clarified by centrifugation at 5.000×g for 1 h, filtered and applied to Glutathione Sepharose 4B beads (GE Healthcare) for 1 h rocking at 4 °C. Beads were washed with RBD buffer (20 mM Tris pH 7.5, 150 mM NaCl, 1 mM

DTT, 1% Triton X-100, 5 mM MgCl2 and 0.05 M PMSF). After the final wash, beads were resuspended in RBD buffer supplemented with 10% glycerol, flash frozen in liquid nitrogen and stored at −80 °C in single-use aliquots. The GST-Rhotekin RBD purification protocol is adapted from ref. [38].

**Crystallization/data collection and structure determination.** Initial small needle clusters of *X. laevis* p190RhoGAP-A pG1 crystals were obtained by sparse matrix screening using a TTP Labtech Mosquito by vapor diffusion in sitting drops at room temperature with a 1:1 (v/v) ratio of purified protein to reservoir solution containing 1.7 M lithium sulfate, 0.1 M sodium acetate pH 4.5. The pH/buffer screening was then carried out using the Slice Screen (Hampton Research), which yielded improvement in crystal singularity and size with substitution of N-(2-Acetamido) iminodiacetic acid (ADA) at pH 6.3 in 1.7–1.8 M lithium sulfate. Crystals were harvested from the drop, quickly incubated in 3.4 M sodium malonate pH 6.3 as a cryoprotectant and flash-cooled in liquid nitrogen. Three sets of diffraction data were collected from a single crystal at Northeastern Collaborative Access Team (NE-CAT) Beamline 24-ID-E at Argonne National Laboratory Advanced Photon Source, processed separately in HKL2000[39], and scaled together in SCALEPACK[39] to improve completeness and resolution to 1.9 Å. The data were initially processed in P6, and PhenixXtriage indicated a point group of P622. Matthews probability calculation indicated a single copy of pG1 in the asymmetric unit for P622. For molecular replacement, initial search models were selected according to HHpred[34, 35] homology detection, which predicted a Ras-like GTPase fold for the p190RhoGAP-A pG1 sequence. Many molecular replacement runs in Phenix Phaser[40, 41] using multiple small GTPase structures as search models failed to discover a structure solution. Next, a more robust search model preparation was performed in mr_rosetta in Phenix[42], which generates search models based on HHpred sequence alignments with homologous structures. A top scoring HHpred homology alignment was that of the small GTPase-like domain of human Arf-GAP with GTPase, Ankyrin repeat and PH domain-containing protein 3 (AGAP3; PDB

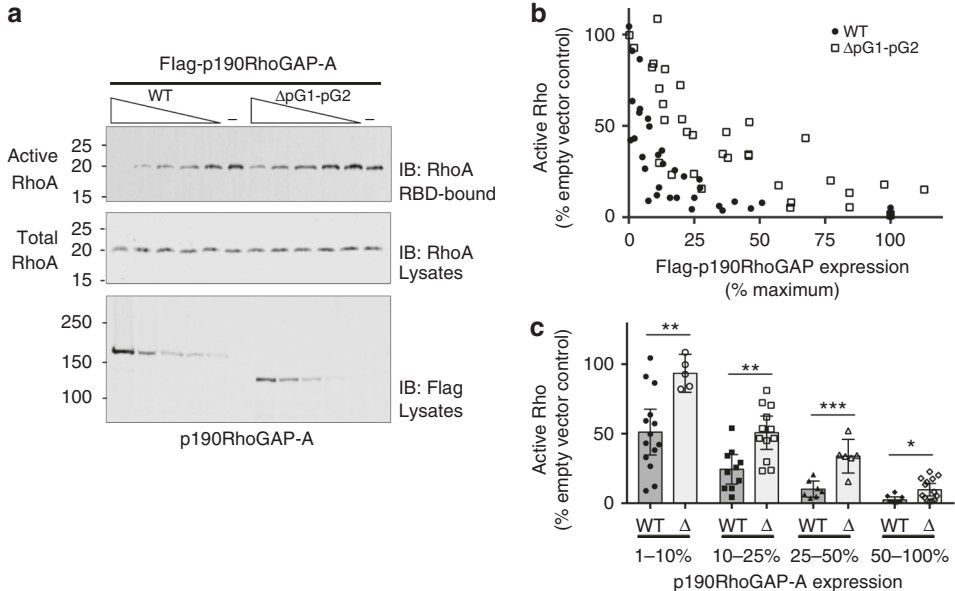

**Fig. 5** Rho activation assays. **a** Active endogenous RhoA levels assessed by pulldown with a GST-rhotekin Rho binding domain (GST-RBD) from HEK293T cells transfected with a gradient of Flag-p190RhoGAP-A, pG1-pG2 deletion mutant (ΔpG1–pG2), or empty Flag vector (−). A representative set of immunoblots is shown. **b** *Scatter plot* of active RhoA as a function of Flag-p190RhoGAP expression. Active RhoA is normalized to maximum levels in each experiment (empty vector control) and Flag-p190RhoGAP expression as a percentage of maximum expression within each experiment. Shown is the compilation of data from 8 independent experiments. **c** *Scatter/bar plot* of data from (**b**), with data points binned according to Flag-p190RhoGAP expression as indicated (1–10%, 10–25%, 25–50% and 50–100%). The mean and s.e.m. (*error bars*) of each group is shown. Unpaired *t*-tests (two-tailed) were performed in GraphPad Prism, with statistical significance indicated above pairs and *p*-values as follows: 1–10% bin $p = 0.0053$; 10–25% bin $p = 0.0019$; 25–50% bin $p = 0.0006$; and 50–100% bin $p = 0.0133$

---

**Table 2 Classification of pseudoGTPases**

| Pseudoenzyme classification | Pseudokinase examples | PseudoGTPase examples |
|---|---|---|
| (i) No nucleotide binding[a] | VRK3, BIR2, ROR2 | p190RhoGAP-A and -B pG1 and pG2, CENP-M, fungal dynein LIC |
| (ii) Nucleotide binding, but with no catalytic activity | ILK, TYK2 JH2, JAK1 JH2, STRADα, MLKL | Rnd family |
| (iii) Retained catalytic activity | WNK1, CASK, ErbB3, JAK2 JH2, KSR2 | RGK family |

Examples of pseudoGTPases and pseudokinases that fit each classification are indicated
[a]Atypical nucleotide (e.g., ATP) or divalent cation ($Mn^{2+}$, etc.) binding cannot be ruled out for all examples

---

accession code 3IHW, unpublished, Structural Biology Consortium, probability 99.8%, *E*-score 2E−17, 13% identical, 129 residues aligned), which was modified by mr_rosetta to generate an improved search model with 1.9 Å r.m.s.d. over 134 aligned residues compared to the original search model. This mr_rosetta model was further modified manually to remove loops connecting predicted secondary structure elements. Finally, the modified models were used as Phaser search models[40], with the best model yielding a final translation function Z-score (TFZ) of 6.7 for a single copy in space group $P6_522$. Initial building was performed in Phenix AutoBuild[43], which built 121 Cα positions, 85 of which were built in correct location (compared to the final refined model) with 18 of these correctly docked in the sequence (residues 657–674) with $R_{free}$ value of 44.9%. A subsequent round of autobuilding in the ARP/wARP module[44] of CCP4i[45, 46] successfully built 142 residues with 134 correctly docked into sequence for a $R_{free}$ of 34.5%. Manual model building was then carried out in Coot[47], and refinement in Phenix[48] including TLS parameters determined in Phenix. The final model contains 157 residues. PISA analysis of crystallographic interfaces supports that the isolated pG1 domains behave as monomers, consistent with gel filtration chromatography. Two malonate ions and three sodium ions are built in p190RhoGAP-A pG1.

Initial crystals of human p190RhoGAP-B were obtained by sparse matrix screening and grew as needle clusters in 2 M ammonium sulfate (AmSO₄), 5% isopropanol. Crystal size could be improved by optimizing reservoir buffer to 2.2 M AmSO₄ and 1% isopropanol, but diffraction studies indicated that freezing was poor. AmSO₄ was substituted with sodium malonate pH 4.7 and crystals grew larger in the absence of isopropanol. Final crystals were grown in 2.15 M sodium malonate pH 4.7 and cryoprotected in 3.4 M sodium malonate pH 4.7 before freezing in liquid nitrogen for diffraction studies. A 2.6 Å data set was

collected from a single crystal at NECAT beamline 24-ID-E and processed in HKL2000[39]. Matthews probability calculator indicated a single copy of p190RhoGAP-B pG1 in the asymmetric unit. A molecular replacement search model was generated using the nearly completed model of *X. laevis* p190RhoGAP-A pG1, which was modified by sculptor[49] to replace nonidentical residues with the p190RhoGAP-B sequence. A single molecular replacement solution was found with Phaser[40] with a TFZ score of 7.2 in space group $P4_3$. PhaserAutoBuild was carried out[43], which resulted in a model containing 113 residues, of which 87 were correctly docked into p190RhoGAP-B sequence, with $R_{free}$ value of 31.9%. Manual model building was carried out in Coot[47], and refinement in Phenix[48] and Refmac5[46] including TLS parameters determined by TLSMD[50]. The final model contains 136 residues. For both crystal structures, model quality was assessed in Molprobity[51]. Crystallographic software is compiled by SBGrid[52].

**Nucleotide-binding assays**. MANT-GTPγS was purchased from Jena Bioscience. Binding of 0.5 μM MANT-GTPγS to 1–2 μM purified protein was measured at room temperature in buffer containing 20 mM Tris pH 7.5, 50 mM NaCl, 10 mM divalent salt ($MgCl_2$, $MnCl_2$, $CaCl_2$, $ZnCl_2$, $NiCl_2$, $CoCl_2$) plus or minus 20 mM EDTA in 100 μl reaction volumes in a black-bottomed microplate. Fluorescence data were collected at excitation and emission wavelengths of 360 and 440 nm, respectively, on a TECAN Infinite M1000 plate reader, or on a BioTek Synergy 2 multi-mode reader with excitation/emission filters of 360/40 and 450/50, respectively. A time course of binding was carried out with fluorescence measurements taken every 30 s for 30 min, with protein added to nucleotide after 3 baseline measurements (which were averaged as signal at time zero). In each experiment,

the fluorescence signal was normalized to time zero. Purified wild-type Rac1 was used as a positive control[9].

Thermal shift assays were performed as described previously for pseudokinases[10]. Protein at 2–5 µM in 20 mM Tris pH 7.4 and 150 mM NaCl was mixed with SYPRO Orange (ThermoFisher; stock solution at 5000×) at a final concentration of 2×–5×, in the absence or presence of 1 mM MgCl₂ or MnCl₂, and in the absence or presence of 200 µM nucleotide: GTP, GTPγS, GDP, GMP or ATP in a total reaction volume of 25 µl. A Bio-Rad CFX Connect Real-time PCR machine with FAM filters was used to collect data. The mixture was pre-equilibrated to 4 °C for 5 min, followed by thermal ramping of 1 °C per min from 4 to 95 °C, with fluorescene measurements taken after each 1 °C increment. Fluorescence signal was normalized and plotted as a function of temperature, and data were fit to a sigmoidal curve in Prism 7 (Graphpad) with $R^2$ values of >0.99. The midway inflection point of the curve represents the melting temperature ($T_m$). Points after the fluorescence maximum were excluded from fitting. Changes in the melting temperature ($\Delta T_m$) compared to the buffer-only control curve were calculated for each ligand pair and reported as the difference in $T_m$. The mean and s.e.m. of three separate experiments was determined. Positive $\Delta T_m$ of ≥3 °C is generally accepted as a significant stabilization of the protein in the presence of ligand[10].

**Rho activity assays**. To measure levels of active Rho[53], HEK293T cells (ATCC) were plated in 6-well culture dishes with Dulbecco's modified Eagle's medium containing 10% fetal bovine serum and penicillin/streptomycin. Cells were transfected with a gradient of pCMV-derived plasmid DNAs from 0.1–5 µg encoding Flag-p190RhoGAP-A wild-type and mutant or empty Flag vector using Lipofectamine 2000 and OPTIMEM (Thermo Fisher). At 68–72 h after transfection, the cells were harvested. Rhotekin pulldown assays were used to assay active Rho in transfected HEK293T cell lysates similarly as described previously[54, 55]. Cells were transferred to ice, washed with ice-cold Tris-buffered saline buffer and lysed in buffer containing 1% v/v NP-40, 25 mM HEPES pH 7.5, 150 mM NaCl, 10 mM MgCl₂, 10% glycerol, 1 mM PMSF, 0.2 mM Na₃VO₄, 10 mM NaF and complete protease inhibitor tablet EDTA-free (Roche). Lysates were clarified by centrifugation at 16,900×g at 4 °C for 5 min and transferred to tubes containing 5 µl of Rhotekin-RBD beads which were obtained commercially from Cytoskeleton, Inc. or prepared as previously described[38, 53]. Lysates were mixed with beads by rocking for 45 min at 4 °C. RBD beads were then washed three times with cold lysis buffer. Bound proteins were eluted with addition of sodium dodecyl sulfate–polyacrylamide gel electrophoresis (SDS–PAGE) sample buffer, resolved by SDS–PAGE and transferred to polyvinylidene difluoride membranes. Bound and total endogenous RhoA was probed by anti-RhoA monoclonal antibody 26C4 at 1:500 dilution (Santa Cruz sc-418), Flag-p190RhoGAP levels in lysates were probed with anti-Flag-M2 antibody at 1:3000 dilution (Millipore Sigma F3165). Anti-mouse IRDye 800CW secondary antibody (LI-COR Biotechnology 926-32212) was used at 1:20,000 dilution. Immunoblots were imaged and quantified on an Odyssey CLx Imaging system and ImageStudio Software (LI-COR Biotechnology). Active RhoA levels, as measured by Rhotekin-bound RhoA, were standardized to total RhoA in each sample, and reported as percent normalized to the maximum level of RhoA pulled down (from empty vector control samples). P-values were calculated using the unpaired t-test, two-tailed, in GraphPad Prism, with significance indicated in the figure legend.

**Data availability**. Coordinates and structure factors have been deposited in the Protein Data Bank under accession codes 5U4U and 5U4V. X-ray diffraction images are available online at SBGrid Data Bank[57] doi:10.15785/SBGRID/454 (5U4U) and doi:10.1038/s41467-017-00483-x (5U4V). Other data are available from the corresponding author on reasonable request.

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

## Acknowledgements

Anthony Koleske is thanked for helpful discussions and for providing p190RhoGAP-A cDNA. Anton Bennett is thanked for providing p190RhoGAP-B cDNA. GST-RBD was a gift from Martin Schwartz (Addgene plasmid #15247). Craig Roy and Justin McDonough are thanked for the use of the Tecan plate reader, and Anatoly Kiyatkin and Mark Lemmon for use of the BioTek Synergy 2 plate reader for the MANT assays. Bertrand Simon, Daniel Iwamoto and LeenaKuruvilla are thanked for assistance with the thermal shift assays. Mark Lemmon is also thanked for helpful comments on the manuscript. Rong Zhang, ByungHak Ha, and Alexander Scherer are thanked for technical input. Staff at beamline 24-ID-E (NE-CAT-E) at the Advanced Photon Source, Argonne National Laboratory are thanked. This work is based upon research conducted at the Northeastern Collaborative Access Team beamlines, which are funded by National Institutes of Health grant P41GM103403. This research used resources of the Advanced Photon Source, a U.S. Department of Energy (DOE) Office of Science User Facility operated for the DOE Office of Science by Argonne National Laboratory under Contract No. DE-AC02-06CH11357. National Institutes of Health grants R01NS085078, R01GM109487, R01GM114621, R01GM102262, R01GM100411, P50CA121974, S10OD018007 funded the research.

## Author contributions

A.L.S. and T.J.B. conceived the project, designed the experiments and wrote the manuscript. A.L.S. conducted the experiments.

## Additional information

**Competing interests:** The authors declare no competing financial interests.

