## [Peer Review File · Nature Communications]

Reviewers' comments:

Reviewer #1 (Remarks to the Author):

The study by Stiegler and Boggon builds on a few previous pieces of work that noted unusual sequence variations in GTP-binding domains (GTPase domains) of proteins, which share the ability to bind (or not!) GTP. The work is strong, timely and well written, and contains an appropriate balance of bioinformatic, structural and cellular studies. In particular, X-ray studies with putative pseudo GTPase domains previously suggested to be unstructured (renamed, and termed pG1 and pG2 here) demonstrate that despite a conserved overall fold, it is unlikely that guanine could bind through a lack of appropriate charged interactions and a potential blocking of the GTP site. Further fluorescence-based data (see below) support a lack of GTP binding (though this assay is prone to error and artefact in other studies), which are combined with an acellular rhotekin pull-down analysis to suggest that RhoGAP activity is maintained when compared to RhoA (although the comparison is made with only one concentration of protein). Finally, pseudoGTPases are then compared across different phyla, to confirm that various types of pseudoenzyme that could correspond to a pseudoGTPase classification are likely exist in this large and varied family.

Major points;

Figure 2: Would it be sensible to compare p190RhoGAP pG1 domain with another small GTPase, perhaps including one lacking a GTP analogue?

In Figure 4a, a lack of binding of MANT-GTPgammaS binding to various G1 domains is apparent. These experiments should be repeated with divalent cations, assuming the assay can tolerate this, and a thermal denaturation approach could be tested, so that non fluorescent GTP (and GMP/GDP/ATP) can be compared, and perhaps also some GTP analogues or small molecules

In Figure 4b, the linearity of the assay needs to be established by comparing different amounts of starting material assayed in the active Rho assay. Currently, the assay might be saturated in terms of Rho activation by very large amounts of p190 and its variants.

Minor points:

In Figure 1A/Supp Fig S1, the Jpred/HHPred profiles for other Ras relatives might be shown.

Three references might be added to bring the breadth of experimental and comparative pseudoenzyme analysis up-to-date,
Scheeff et al., Structure 2009; 17
Murphy et al., Biochem J 2014 457;
Reiterer et al., TICB 2014 24;

In Figure 5, it is possible that other nucleotides bind to pseudoenzymes, so it is worth pointing out that only ATP/GTP have been tested (in vitro) in the legend. Worth stating also that the effects of cations (or other biochemicals) might be relevant.

Reviewer #2 (Remarks to the Author):

Domain assignment searches led to the identification of two putative "GTPase-like" domains (called pG1 and pG2) in the middle regions of the RHOGAPs p190A and p190B. The crystal structures of pG1 domains p190A and p190B (diffract to 1.9 Å and 2.6 Å resolution, respectively) consist of a central 6-stranded β-sheet surrounded by four α-helices. The RMSDs between the two pG1 structures is 1.8 Å over 128 Ca's and 2.7 Å over 135 Ca's between HRAS and p190A pG1. Structure and sequence homology analysis showed that the guanine nucleotide binding motifs (G1-G5) are almost completely absent. So no bound nucleotide was detected in the pG1 structures.

Efforts, such as mant-GTP γ S binding in a fluorescence assay or co-crystallization of pG1 with GTP γ S, did not result in nucleotide binding by pG1. This and the lack of G-motif conservation led to the classification of p190 pG1 and pG2 as pseudoGTPase domains. A functional consequence of pG1 and pG2 deletion on p190 RHOGAP activity, analyzed by overexpression in HEK293T cells and RHOA-GTP pulldown, remained obscure. As pG1/pG2 function(s) remained unclear it is generally concluded that these newly identified domains may provide sites for new interactions in signal transduction as previously proposed for pseudokinases with scaffolding/adaptor functions.

Major points:

This manuscript is clear and informative but in some parts too short.

1) It would be very helpful for readers to describe some parts in more details how precisely did the authors identify those pG1 and pG2 domains? I took my time to repeat such search and could reproduce it after several re-setups and re-submissions.

2)) There are very likely pG1 and pG2 domains in both p190A and p190B but it is not clear (or not shown: line 60) how thoroughly authors considered their presence in diverse species. The strategy to use proteins from many species is elegant and dramatically increases a chance to achieve relevant results as the presented study shows. Just for the complete picture it should be discussed if features found for some of them are valid in general as stated in the abstract of the paper (line 26). pG2 is in fact not proven as authors did not succeed to obtain the structure of pG2 and its existence should be thus supported by the means of such discussion.

3) The common principle is that 160-180 residues of G domain with an alpha-beta topology is responsible for nucleotide-dependent conformational changes and drives many biological functions (doi: 10.1146/annurev-biochem-062708-134043). Even the mutation of only one residues in RAS (Asp to Asn in the NKXD motif) changes GDP binding characteristics to XDP (Oncogene 1996 12:87-96). Other very modest sequence deviations can significantly change the biochemistry of proteins that tightly bind and hydrolyze GTP (doi: 10.1515/hsz-2012-0207). pG1 and pG2 even lack almost all signatures (Tab. S3) which determine specific binding of guanine nucleotides, as the author stated (lines 86-87): "Together, the substantially degraded G-motifs in the p190RhoGAP pG1 domain indicate an inability to bind nucleotide." In addition, RAS superfamily proteins adopt a typical Rossmann fold topology, which is found in very many proteins beyond the RAS superfamily. Thus, it is very difficult to understand why pG1 and/or pG2 were called "pseudoGTPase" because they may also bind other types of nucleotides. It is of major importance to discuss this central point of the manuscript.

4) It is difficult to comprehend and remained unclear why pG1 is structurally compared to HRAS (Fig. 2 and Fig. 3) but biochemically compared to RAC in terms of nucleotide binding (Fig. 4a). Even more confusing it is in relation to Tab. S1 that provides information about the closest homologues and comprises mostly Rab GTPases. Figure S3 compares accordingly predicted secondary structure of pG2 with RAB8A but other relations to proteins from this subfamily is missing. Readers would benefit a lot from a sequence alignment in the style of Table S3 that would also include relevant/representative GTPases from the Table S1 accompanied with corresponding structural comparisons.

5) Is there any reason why the N-terminal GTPase domain in p190 protein remained unmentioned. Is there any similarity or even relation between this poorly analyzed domain and the pG1/pG2 domains!

6) In the end, identification of two additional domains in the RHOGAP proteins p190A and p190B is very interesting but also very disappointing: The role of the pG1/pG2 integrity and activity in p190 function remains unclear.

Minor point:

1) Introduction (line 42): Many readers might not know much about pseudokinases or

pseudoGTPases; so, the authors could better introduce some points at this position of the manuscript instead later (lines 107 to 112).

2) Purification of pG1 domains have been performed in the absence of GDP, which is known to result in nucleotide depletion and inactivation of many GTPases. It is suggested to purify pG1 and/or pG2 in the presence of GDP (or any other nucleotide or nucleotide-like cofactors) in the lysis buffer before purification by chromatography, dialysis, etc..

3) Figure 4b: in the control experiment (empty vector) the authors claim around 10 % RHO activation! They do not mention how they have quantified this! Probably, this should not be called quantification by percentage instead by normalization to 1.

4) Figure 5 looks like a table.

5) Figure S4 describes conservation analysis of pG1. But conservation related to what remains unclear. An alignment may be very helpful.

6) Table S3: The second glycine in the P-loop (G12 in HRAS), essential for the intrinsic and GAP-induced GTP hydrolysis, should be highlighted.

Reviewer #3 (Remarks to the Author):

The paper reports the identification and structural determination of pseudo-GTPases within a central region of two related p190RhoGAPs (A and B). RhoGAPs regulate Rho-mediated cytoskeletal dynamics, and the structurally characterized domain from p190RhoGAP-A is the first of two tandem 'cryptic' pseudo-GTPases that had not previously been identified. The authors performed extensive secondary structure and domain prediction analyses, using software tools that are available online. The crystal structures of the first pseudo-GTPase domain (G1) from *Xenopus* (RhoGAP-A) and human (RhoGAP-B) are described in this paper.

Although the authors have shown experimentally that these cryptic domains are indeed incapable of acting like canonical Ras-like GTPases, there is currently no functional relevance attributed to these domains in p190RhoGAP. It is essential to connect the structural findings to the biological context of Rho-dependent cytoskeletal dynamics. Additionally, it has already been shown that 'pseudo-GTPases exist in mammalian cytosolic signaling cascades' (quote from page 5, bottom), for example the kinesin regulator DLIC from the Vale lab. Although it was a thermophilic organism, certainly the implication is that pseudo-GTPases regulate cytoplasmic cargo trafficking. Therefore, the findings reported here provide additional archival data on structures of pseudo-GTPases, rather than novel conceptual advances that shed light on their functions.

Finally, comparisons of the non-canonical small GTPases with the pseudo-kinase family of proteins seem premature, given that only a handful of these GTPases have been identified to date. Can the authors provide evidence (sequence signatures, homology predictions, etc) from a genome-wide analyses that these pseudo-GTPases are widespread? Along with functional data, this would be an impressive finding, relevant to the wider scientific community.

Here are a few other comments and suggestions:

- constructs encompassing the tandem domains together (both pseudo-GTPases) were not soluble. It would be interesting and more physiologically relevant to understand the structure/function of this larger fragment, since they could act together to regulate the RhoGAP. On page 3, the corresponding sequences in one of the RhoGAPs is 592-767 (G1) and 779-950 (G2) - authors should indicate here, and in supplementary which RhoGAP they are referring to (A vs. B, human?).

Therefore, only 10 residues link G1 (alpha-helix 5) and G2 (b-strand 1), so there is a good chance the two domains work together as a structural unit. It would be a priority in future experiments to understand, biophysically, structurally and functionally, the tandem unit.

- the line 'PISA analysis does not suggest complex formation' is a little vague (p. 5). Are the authors implying that PISA analysis indicates the protein is a monomer? The asymmetric unit is a monomer in both structures of G1 domain (RhoGAP A and B), and a monomeric solution state could easily be confirmed using static light scattering or analytical ultracentrifugation, at least for the isolated domain. Alternatively, do the authors mean that there are no predicted 'protein-protein' interaction regions predicted by PISA?

- how similar in sequence are the G1 and G2 domains among the mammalian RhoGAPs? It is noted that several constructs were tested for expression and crystallization (rat, human, etc) and it would be useful to know the sequence identities, relative conservation of motifs, etc, between mammalian proteins. If there are structural/functional features that are conserved in the mammalian 'sub-family' of pseudo-GTPases, it is possible that sequence alignments would possibly bring these features out.

- I'm not sure how useful comparisons between canonical Ras motifs and pseudo-GTPases are, except to emphasize the point that they are 'degraded' (or non-conserved). It would be interesting, however, to compare more deeply the known pseudo-enzymes and try to find common threads that link the structure to their alternative functions. Unfortunately, as the authors discuss, there are few known examples and it may be too speculative at this point.

- Figure 1 is more appropriate for Supplementary, or simply references to the web programs, since standard predictions using online software have been used in this instance

In summary, the authors have taken an important step forward with the identification of an unknown set of tandem domains in p190RhoGAP, and their structural characterization. However, in the absence of a functional relevance for these domains, the crystal structures of the G1 domains do not contribute significantly to our understanding of Rho-mediated cell dynamics.

Responses to reviewers' comments:

Reviewer #1 (Remarks to the Author):

The study by Stiegler and Boggon builds on a few previous pieces of work that noted unusual sequence variations in GTP-binding domains (GTPase domains) of proteins, which share the ability to bind (or not!) GTP. The work is strong, timely and well written, and contains an appropriate balance of bioinformatic, structural and cellular studies. In particular, X-ray studies with putative pseudo GTPase domains previously suggested to be unstructured (renamed, and termed pG1 and pG2 here) demonstrate that despite a conserved overall fold, it is unlikely that guanine could bind through a lack of appropriate charged interactions and a potential blocking of the GTP site. Further fluorescence-based data (see below) support a lack of GTP binding (though this assay is prone to error and artefact in other studies), which are combined with an acellular rhotekin pull-down analysis to suggest that RhoGAP activity is maintained when compared to RhoA (although the comparison is made with only one concentration of protein). Finally, pseudoGTPases are then compared across different phyla, to confirm that various types of pseudoenzyme that could correspond to a pseudoGTPase classification are likely exist in this large and varied family.

We thank reviewer #1 for the very favorable remarks that the study is “strong, timely and well written” and that our work “contains an appropriate balance of bioinformation, structural and cellular studies.” We have addressed the specific concerns below.

Major points;

Figure 2: Would it be sensible to compare p190RhoGAP pG1 domain with another small GTPase, perhaps including one lacking a GTP analogue?

The only crystal structures of small GTPases lacking nucleotide in the binding site are in complex with their cognate GEF (e.g., H-Ras with Sos). The lack of nucleotide-free structures of small GTPases is most likely due to the instability of these proteins without their ligand. So, although this is an intriguing question (comparison of our structures with the conformation of other small GTPases in the absence of bound nucleotide), we don't think the current composition of the protein data bank allows a meaningful analysis.

In Figure 4a, a lack of binding of MANT-GTPgammaS binding to various G1 domains is apparent. These experiments should be repeated with divalent cations, assuming the assay can tolerate this...

This is a good suggestion, and we have directly addressed it by testing other divalent cations in the MANT-GTPgammaS binding assay. We analyzed a panel of 6 divalent cations (Mg, Mn, Ca, Zn, Ni, Co), in the absence and presence of EDTA to promote nucleotide exchange. Again, we find no signal corresponding to binding of MANT-GTPgammaS. These new data are included in Figure 4c.

...and a thermal denaturation approach could be tested, so that non fluorescent GTP (and GMP/GDP/ATP) can be compared, and perhaps also some GTP analogues or small molecules

This is another good suggestion. As per the Reviewer 1's comments, we have performed thermal denaturation assays using SYPRO Orange protein dye, similar to those studies performed for a large subset of pseudokinases by Murphy et. al. , Biochemical Journal 2014. We use wild-type Rac1 as a positive control, and observe that Rac1 is stabilized by divalent cations (Mg and Mn), nucleotide (GTP/GDP/GTPgS but not GMP or ATP), and additively by both, reflecting its ability to bind both divalent and nucleotide (Figure 4f,i). In contrast, the melting temperature of p190RhoGAP-A and -B pG1 domains is not changed in the presence of divalent, nucleotide, or both (Figure 4d,e,g,h). These results provide an orthogonal

study that confirms our MANT binding assay findings. We believe addition of this analysis makes this section of the study stronger, and we thank the reviewer the comment.

In Figure 4b, the linearity of the assay needs to be established by comparing different amounts of starting material assayed in the active Rho assay . . . (the comparison is made with only one concentration of protein). Currently, the assay might be saturated in terms of Rho activation by very large amounts of p190 and its variants.

This is another excellent point. To address this we have improved the Rhotekin binding assay by transfecting a gradient of plasmid DNA, and by assessing endogenous RhoA. We are confident that we have achieved “linearity” in the assay, present our new data in a new figure (Figure 5), and briefly describe the improvements to the assay below.

Gradient of plasmid DNA. We now achieve an expression curve of Flag-tagged p190RhoGAP-A in HEK293T cells by transfecting a gradient of plasmid DNA. We observe a clear dose response for both p190RhoGAP-A constructs used.

Assessing endogenous RhoA. We now assess activity of transfected p190RhoGAP-A on endogenous RhoA rather than co-transfecting exogenous RhoA. We find that the response is proportional to expression levels of p190RhoGAP-A.

Reassessment of original assays. In light of these new data we re-analyzed our original experiments. We now believe that these were most likely at saturating levels of Flag-p190RhoGAP-A expression leading to low levels of active RhoA. For clarity we have replaced the original assays with our new ones.

These reviewer-inspired improvements to our Rhotekin assay have allowed us to observe significant differences in cellular levels of active RhoA when comparing wild-type or mutant (deletion of pG1-pG2) p190RhoGAP-A expression. We find that deletion of pG1-pG2 leads to higher levels of active RhoA in the cell compared to wild-type p190RhoGAP-A at comparable expression levels. The combination of titration of p190RhoGAP-A expression alongside assessment of endogenous RhoA has significantly improved this assay. We believe that this new, and exciting, observation lays the groundwork for future studies to better understand p190RhoGAP regulation.

Minor points:

In Figure 1A/Supp Fig S1, the Jpred/HHPRed profiles for other Ras relatives might be shown.

In Supplementary Figure S1, we have now included the DSSP secondary structure profiles from additional members of the Ras superfamily.

Three references might be added to bring the breadth of experimental and comparative pseudoenzyme analysis up-to-date, Scheeff et al., Structure 2009 17; Murphy et al., Biochem J 2014 457; Reiterer et al., TICB 2014 24.

These references are now included in the manuscript (16, 15, 27, respectively).

In Figure 5, it is possible that other nucleotides bind to pseudoenzymes, so it is worth pointing out that only ATP/GTP have been tested (in vitro) in the legend. Worth stating also that the effects of cations (or other biochemicals) might be relevant.

Figure 5 is now Table 1. We have amended the legend to reflect this concern.

Reviewer #2 (Remarks to the Author):

Domain assignment searches led to the identification of two putative “GTPase-like” domains (called pG1 and pG2) in the middle regions of the RHOGAPs p190A and p190B. The crystal structures of pG1 domains p190A and p190B (diffract to 1.9 Å and 2.6 Å resolution,

respectively) consist of a central 6-stranded β -sheet surrounded by four α -helices. The RMSDs between the two pG1 structures is 1.8 Å over 128 Ca's and 2.7 Å over 135 Ca's between HRAS and p190A pG1. Structure and sequence homology analysis showed that the guanine nucleotide binding motifs (G1-G5) are almost completely absent. So no bound nucleotide was detected in the pG1 structures. Efforts, such as mant-GTP γ S binding in a fluorescence assay or co-crystallization of pG1 with GTP γ S, did not result in nucleotide binding by pG1. This and the lack of G-motif conservation led to the classification of p190 pG1 and pG2 as pseudoGTPase domains. A functional consequence of pG1 and pG2 deletion on p190 RHOGAP activity, analyzed by overexpression in HEK293T cells and RHOA-GTP pulldown, remained obscure. As pG1/pG2 function(s) remained unclear it is generally concluded that these newly identified domains may provide sites for new interactions in signal transduction as previously proposed for pseudokinases with scaffolding/adaptor functions.

Major points:

This manuscript is clear and informative but in some parts too short.

We thank the reviewer for their kind words, and have expanded the text. We have addressed their specific comments below.

1) It would be very helpful for readers to describe some parts in more details how precisely did the authors identify those pG1 and pG2 domains? I took my time to repeat such search and could reproduce it after several re-setups and re-submissions.

We have edited both the main text (p. 3) and the Supplementary methods (p. 2) to include a more detailed, step-by-step procedure for our initial identification of pG1 and pG2 by bioinformatics.

2) There are very likely pG1 and pG2 domains in both p190A and p190B but it is not clear (or not shown: line 60) how thoroughly authors considered their presence in diverse species. The strategy to use proteins from many species is elegant and dramatically increases a chance to achieve relevant results as the presented study shows. Just for the complete picture it should be discussed if features found for some of them are valid in general as stated in the abstract of the paper (line 26).

We agree with the reviewer. As they point out, our previous version did not adequately describe our deep search of sequence databases for p190RhoGAP proteins and evaluation for the presence of pG1 and pG2. We now include improved discussion of our searches in the text (p. 4). We also expand Supplementary Table S1 to show that these domains are conserved in both -A and -B p190RhoGAP homologs in vertebrates, in the single p190RhoGAP protein in insects (*Drosophila melanogaster*), and in sponge (*Amphimedon queenslandica*) which also has a single p190RhoGAP. In our updated table we show that HHpred predicts GTPase-like folds in the middle domains of each of these proteins despite the sequence conservation between human and sponge of only ~22%, for example. The top HHpred results are listed in Supplementary Table S1.

pG2 is in fact not proven as authors did not succeed to obtain the structure of pG2 and its existence should be thus supported by the means of such discussion.

We agree to some extent that it was more challenging to make the case that pG2 is a pseudoGTPase like pG1, since we do not currently have three-dimensional data to directly support this idea. However, the consistent homology predictions of pG2 from p190RhoGAP-A and -B from vertebrates and also from lower species like *Drosophila* and even sponge is quite compelling (Supplementary Table S1). Furthermore, the high degree of certainty with which this prediction is made raises the likelihood that this region folds similar to a GTPase domain (Supplementary Table S1). When considering pG2 as a 'pseudo'GTPase, we used homology-

based alignment (taking into account secondary structure alignment) which we now include in an expanded Supplementary Figure S3. This figure now highlights that the putative G motifs of pG2, which have been identified by multiple sequence and homology alignment, are highly degraded in sequence. Accordingly, we discuss these searches and alignments in the main text on p. 5.

3) The common principle is that 160-180 residues of G domain with an alpha-beta topology is responsible for nucleotide-dependent conformational changes and drives many biological functions (doi: 10.1146/annurev-biochem-062708-134043). Even the mutation of only one residues in RAS (Asp to Asn in the NKXD motif) changes GDP binding characteristics to XDP (Oncogene 1996 12:87-96). Other very modest sequence deviations can significantly change the biochemistry of proteins that tightly bind and hydrolyze GTP (doi: 10.1515/hsz-2012-0207). pG1 and pG2 even lack almost all signatures (Tab. S3) which determine specific binding of guanine nucleotides, as the author stated (lines 86-87): "Together, the substantially degraded G-motifs in the p190RhoGAP pG1 domain indicate an inability to bind nucleotide." In addition, RAS superfamily proteins adopt a typical Rossmann fold topology, which is found in very many proteins beyond the RAS superfamily. Thus, it is very difficult to understand why pG1 and/or pG2 were called "pseudoGTPase" because they may also bind other types of nucleotides. It is of major importance to discuss this central point of the manuscript.

To directly address the possibility that pG1 is an atypical GTPase in its binding specificity, (also raised by reviewer #1), we performed further biochemical studies. Using a thermal shift assay, we assessed binding to ATP and GMP (in addition to GTP, GDP and GTPyS) and divalent cations Mg^{2+} and Mn^{2+} (Figure 4). We observe no stabilization of pG1 protein in these conditions, strongly suggesting that pG1 does not bind these atypical ligands. We also repeated the MANT-GTPyS fluorescence binding assay in the presence of a larger panel of atypical divalent cations (Mn, Ca, Zn, Ni, Co), and find that pG1 does not bind MANT-GTPyS in these conditions. So, although we agree with the reviewer that this is a Rossmann fold, we believe that the evidence of the structure analysis and nucleotide binding studies strongly suggest that this is a bone-fide 'pseudoGTPase' domain.

4) It is difficult to comprehend and remained unclear why pG1 is structurally compared to HRAS (Fig. 2 and Fig. 3) but biochemically compared to RAC in terms of nucleotide binding (Fig. 4a).

We did not intend to confuse the readers by comparing pG1 to both H-Ras and Rac1 in different parts of the manuscript, but rather use both examples in different settings for practical purposes. First, we used the structure of H-Ras as the prototypical small GTPase for structural comparisons because this will be readily identified by a large audience. Second, we have measured GTP binding to Rac1 in MANT fluorescence assays in previous studies (Davis M et al 2013 PNAS), and therefore used Rac1 as a positive control.

Even more confusing it is in relation to Tab. S1 that provides information about the closest homologues and comprises mostly Rab GTPases. Figure S3 compares accordingly predicted secondary structure of pG2 with RAB8A but other relations to proteins from this subfamily is missing. Readers would benefit a lot from a sequence alignment in the style of Table S3 that would also include relevant/representative GTPases from the Table S1 accompanied with corresponding structural comparisons.

We have made the suggested changes to Supplementary Table S1 and Supplementary Figure S3, as discussed above (point 2). We hope these changes sufficiently address the reviewer's concerns over the representation of pG2 as a pseudoGTPase.

5) Is there any reason why the N-terminal GTPase domain in p190 protein remained unmentioned. Is there any similarity or even relation between this poorly analyzed domain and the pG1/pG2 domains!

The reviewer raises an interesting question. We excluded discussion about the N-terminal GTPase domain because it appears largely unrelated to the newly discovered pG1 and pG2 domains. p190RhoGAP-A pG1 and pG2 share little sequence similarity with the N-terminal GTPase domain: only 12% and 21% identity, respectively. The N-terminal GTPase domain further distinguishes itself from these pseudoGTPase domains in that it reportedly binds GTP. As per the reviewer's comments, we have included a brief comparison of pG1 and pG2 with the N-terminal GTPase domain in the manuscript, but feel a more detailed comparison is not relevant to the present story.

6) In the end, identification of two additional domains in the RHOGAP proteins p190A and p190B is very interesting but also very disappointing: The role of the pG1/pG2 integrity and activity in p190 function remains unclear.

We too were initially disappointed at the apparent lack of functional relevance of these newly identified domains in p190RhoGAP. However, at the request of reviewer #1, we now include new and improved Rhotekin pulldown assays (Figure 5) which have uncovered an unanticipated increase in levels of active RhoA when the pG1-pG2 region is deleted from p190RhoGAP. This suggests this region may contribute to full GAP activity of p190 toward Rho. While the exact mechanism is not understood, we are excited by this finding and believe it adds depth to our study and sets the groundwork for future studies.

Minor point:

1) Introduction (line 42): Many readers might not know much about pseudokinases or pseudoGTPases; so, the authors could better introduce some points at this position of the manuscript instead later (lines 107 to 112).

We thank the reviewer for this idea and agree. Thus, we have included a more detailed description of pseudokinases in the opening paragraph. We hope that introducing this information earlier in the manuscript aids the reader.

2) Purification of pG1 domains have been performed in the absence of GDP, which is known to result in nucleotide depletion and inactivation of many GTPases. It is suggested to purify pG1 and/or pG2 in the presence of GDP (or any other nucleotide or nucleotide-like cofactors) in the lysis buffer before purification by chromatography, dialysis, etc..

We are confident in our conclusion that pG1 is not a nucleotide and/or divalent cation binding domain. Our resubmission includes further assessment of in vitro nucleotide and divalent cation binding by pG1 in the fluorescence-MANT assay (Figure 4 a-c) and new thermal shift assays (Figure 4 d-i). Therefore, we do not expect that purification in the presence of GDP will improve the yield of pG1 or pG2 (which we believe is insoluble for reasons independent of nucleotide).

3) Figure 4b: in the control experiment (empty vector) the authors claim around 10 % RHO activation! They do not mention how they have quantified this! Probably, this should not be called quantification by percentage instead by normalization to 1.

Our Rhotekin pulldown assays are now shown in Figure 5. We now quantify the % active RhoA on a normalized scale from 0-100% (100% being the maximum level of active RhoA from an empty vector control lane) in these updated experiments. We hope the reviewer agrees that this is a more suitable way to present the data.

4) Figure 5 looks like a table.

This is now Table 1.

5) Figure S4 describes conservation analysis of pG1. But conservation related to what remains unclear. An alignment may be very helpful.

The surface conservation analysis was carried out using the pG1 domains from p190RhoGAP-A proteins from 74 manually curated sequences. We now include the alignment used for ConSurf analysis as part c of Supplementary Figure S4.

6) Table S3: The second glycine in the P-loop (G12 in HRAS), essential for the intrinsic and GAP-induced GTP hydrolysis, should be highlighted.

We agree with the reviewer that the G12 equivalent position is worth including in the P-loop motif and have changed Table S3 accordingly.

Reviewer #3 (Remarks to the Author):

The paper reports the identification and structural determination of pseudo-GTPases within a central region of two related p190RhoGAPs (A and B). RhoGAPs regulate Rho-mediated cytoskeletal dynamics, and the structurally characterized domain from p190RhoGAP-A is the first of two tandem 'cryptic' pseudo-GTPases that had not previously been identified. The authors performed extensive secondary structure and domain prediction analyses, using software tools that are available online. The crystal structures of the first pseudo-GTPase domain (G1) from *Xenopus* (RhoGAP-A) and human (RhoGAP-B) are described in this paper.

We thank reviewer #3 for their thoughtful and careful comments on our manuscript, and address their specific comments below.

Although the authors have shown experimentally that these cryptic domains are indeed incapable of acting like canonical Ras-like GTPases, there is currently no functional relevance attributed to these domains in p190RhoGAP. It is essential to connect the structural findings to the biological context of Rho-dependent cytoskeletal dynamics.

As described below, and in our rebuttal to reviewer #1, improvements to our Rhotekin pull-down assays have allowed us to show a functional impact of these domains.

Additionally, it has already been shown that 'pseudo-GTPases exist in mammalian cytosolic signaling cascades' (quote from page 5, bottom), for example the kinesin regulator DLIC from the Vale lab. Although it was a thermophilic organism, certainly the implication is that pseudo-GTPases regulate cytoplasmic cargo trafficking. Therefore, the findings reported here provide additional archival data on structures of pseudo-GTPases, rather than novel conceptual advances that shed light on their functions.

We respectfully disagree with the assessment that our discovery of two pseudoGTPase domains in p190RhoGAP proteins is merely "archival." While the reviewer is correct that pseudoGTPase domains have been identified in other proteins previously such as DLIC, there are relatively few known pseudoGTPases compared to the larger class of pseudokinases, and the potential roles of pseudoGTPases in signaling are only beginning to be understood. Therefore, discovery of these domains in p190RhoGAP has the potential to shed light on the roles of the growing family of pseudoGTPases as a whole. Furthermore, p190RhoGAP is a critical regulator of Rho function in the cell, and the surprising discovery of these domains will now allow for new approaches in future studies aimed at better determining its function. This is especially true in light of our new findings that pG1-pG2 may contribute to full p190RhoGAP activity toward RhoA (Figure 5).

Finally, comparisons of the non-canonical small GTPases with the pseudo-kinase family of proteins seem premature, given that only a handful of these GTPases have been identified to date. Can the authors provide evidence (sequence signatures, homology predictions, etc) from a genome-wide analyses that these pseudo-GTPases are widespread? Along with functional data, this would be an impressive finding, relevant to the wider scientific community.

We agree that evidence of a more widespread existence of pseudoGTPase folds in the proteome would be a powerful testament to their emerging importance in cellular function. However, the identification of these domains from primary sequence in a bioinformatics search is challenging: they have low primary sequence identity with active small GTPases (in general much less than 20%). We were only able to initially identify these domains by a homology prediction, but a search of this nature for the entire proteome is currently intractable.

Here are a few other comments and suggestions:

- constructs encompassing the tandem domains together (both pseudo-GTPases) were not soluble. It would be interesting and more physiologically relevant to understand the structure/function of this larger fragment, since they could act together to regulate the RhoGAP.

We agree that studies utilizing the larger pG1-pG2 fragment could be more enlightening with respect to their function in p190RhoGAP, but unfortunately tandem pG1-pG2 constructs are insoluble.

On page 3, the corresponding sequences in one of the RhoGAPS is 592-767 (G1) and 779-950 (G2) - authors should indicate here, and in supplementary which RhoGAP they are referring to (A vs. B, human?).

We thank the reviewer for pointing out this omission. We have clarified which p190RhoGAP, species, and numbering is being used.

Therefore, only 10 residues link G1 (alpha-helix 5) and G2 (b-strand 1), so there is a good chance the two domains work together as a structural unit. It would be a priority in future experiments to understand, biophysically, structurally and functionally, the tandem unit.

The approximately 10-12 residue linker that is predicted to connect pG1 with pG2 is, in theory, sufficiently long to allow pG1 and pG2 to behave as “beads on a string;” however, interdomain interactions involving the pseudoGTPase domains and/or linker itself could allow the region to form a more compact unit. We agree that this potential can be addressed in future studies when/if we are able to achieve soluble recombinant protein encompassing this larger region of 190RhoGAP.

- the line ‘PISA analysis does not suggest complex formation’ is a little vague (p. 5). Are the authors implying that PISA analyses indicates the protein is a monomer? The asymmetric unit is a monomer in both structures of G1 domain (RhoGAP A and B), and a monomeric solution state could easily be confirmed using static light scattering or analytical ultracentrifugation, at least for the isolated domain. Alternatively, do the authors mean that there are no predicted ‘protein-protein’ interaction regions predicted by PISA?

We thank the reviewer for pointing out this confusing sentence. We have changed the wording (on p. 5) of the Supplement in order to clarify our intended meaning, which is that PISA analysis of the crystal packing does not indicate the likelihood that the protein forms higher order oligomers (including dimers). This is consistent with our size exclusion chromatography results, which support that pG1 behaves as a monomer in solution.

- how similar in sequence are the G1 and G2 domains among the mammalian RhoGAPs? It is noted that several constructs were tested for expression and crystallization (rat, human, etc) and it would be useful to know the sequence identities, relative conservation of motifs, etc, between mammalian proteins. If there are structural/functional features that are conserved in the mammalian 'sub-family' of pseudo-GTPases, it is possible that sequence alignments would possibly bring these features out.

The pG1 domain and pG2 domain are each relatively conserved among mammalian p190RhoGAP-A or -B. Specifically, the % identities of the pG1 domains which we used for crystallization are included in the Supplemental Methods (p. 2-3). We have also included the sequence alignment that was used to create the surface conservation figure (Fig. S4) which includes 74 sequences of pG1 from p190RhoGAP-A.

- I'm not sure how useful comparisons between canonical Ras motifs and pseudo-GTPases are, except to emphasize the point that they are 'degraded' (or non-conserved). It would be interesting, however, to compare more deeply the known pseudo-enzymes and try to find common threads that link the structure to their alternative functions. Unfortunately, as the authors discuss, there are few known examples and it may be too speculative at this point.

We agree with the reviewer that a deeper investigation of the emerging group of pseudoGTPases at the sequence level could be enlightening, and also agree that currently the sample number is too small to generate a meaningful outcome. For example, when comparing several of the known pseudoGTPases (p190RhoGAP, CENP-M and DLIC), we find they share less than 20% identity, and close examination of their alignment does not indicate any potential functional "hotspot." What is shared is the common core GTPase-like fold and lack of conserved G-motifs.

- Figure 1 is more appropriate for Supplementary, or simply references to the web programs, since standard predictions using online software have been used in this instance

We thank the reviewer for the suggestion, but have decided to leave Figure 1 unchanged, because we feel it aids the reader's understanding of the study.

In summary, the authors have taken an important step forward with the identification of an unknown set of tandem domains in p190RhoGAP, and their structural characterization. However, in the absence of a functional relevance for these domains, the crystal structures of the G1 domains do not contribute significantly to our understanding of Rho-mediated cell dynamics.

We believe that our revised manuscript is significantly improved, and hope that reviewer #3 agrees.

REVIEWERS' COMMENTS:**Reviewer #1 (Remarks to the Author):**

The study now appears very suitable for publication, and explores a really interesting research area (pseudo-GTPases) that is growing rapidly. I am very pleased to say that all my comments have been adequately addressed, and that the DSF studies and re-analysis of the Rhotekin assay reveal clear, and interesting, data. This is a nice piece of work, and I look forward to seeing the paper published.

Pat Evers

Reviewer #2 (Remarks to the Author):

The authors addressed the main concerns from the reviews. The revised version of the manuscript in total appears to be very good.